# Statin Therapy and the Risk of Viral Infection: A Retrospective Population-Based Cohort Study

**DOI:** 10.3390/jcm11195626

**Published:** 2022-09-24

**Authors:** Biing-Ru Wu, Ding-Han Chen, Wei-Chih Liao, Wen-Chao Ho, Ming-Chien Yin, Cheng-Li Lin, Chia-Hui Chou, Yi-Hao Peng

**Affiliations:** 1Division of Pulmonary and Critical Medicine, Department of Internal Medicine, China Medical University Hospital, China Medical University, Taichung 404327, Taiwan; 2Department of Respiratory Therapy, China Medical University Hospital, China Medical University, Taichung 404327, Taiwan; 3Hyperbaric Oxygen Therapy Center, China Medical University Hospital, China Medical University, Taichung 404327, Taiwan; 4School of Medicine, China Medical University, Taichung 404333, Taiwan; 5Department of Public Health, China Medical University, Taichung 406040, Taiwan; 6Department of Nursing, Asia University, Taichung 413305, Taiwan; 7Management Office for Health Data, China Medical University Hospital, Taichung 404327, Taiwan; 8Division of Infectious Diseases, Department of Internal Medicine, China Medical University Hospital, China Medical University, Taichung 404327, Taiwan; 9Department of Respiratory Therapy, Asia University Hospital, Asia University, Taichung 413505, Taiwan

**Keywords:** statin therapy, viral infection, epidemiology, cohort study, propensity score matching

## Abstract

Statins exert cholesterol-independent beneficial effects, including immunomodulatory effects. In this study, we attempted to investigate the association between statin therapy and the risk of viral infection. We conducted a retrospective cohort study using data from Taiwan’s National Health Insurance Research Database. We identified patients with hyperlipidemia and divided them into two cohorts: statin users and statin nonusers. A 1:1 propensity score matching was conducted between the two cohorts, and a Cox proportional hazards model was used to evaluate the risk of viral infection. Overall, a total of 20,202 patients were included in each cohort. The median follow-up durations were 4.41 and 6.90 years for statin nonusers and users, respectively. The risk of viral infection was 0.40-fold (95% confidence interval = 0.38–0.41) in statin users than in statin nonusers after adjustment for potential confounders. Statin treatment was associated with a significantly lower risk of viral infection in all age groups older than 18 years in both men and women. Moreover, the risk of viral infection substantially reduced as the duration of statin treatment increased. Our findings suggest that statin therapy is associated with a significantly lower risk of viral infection in patients with hyperlipidemia.

## 1. Introduction

Statins are safe, inexpensive, and effective inhibitors of cholesterol biosynthesis, and they have been the mainstay treatment for hyperlipidemia and atherosclerotic cardiovascular disease for over 30 years [1,2]. Numerous experimental and clinical studies have demonstrated that statins exert cholesterol-independent beneficial effects; for example, they inhibit oxidative stress and inflammation, stabilize atherosclerotic plaques, and modulate the immune system [3,4,5,6].

Statins regulate the immune response on many levels [5,7]; they exert their effects by inhibiting 3-hydroxy-3-methyl-glutaryl-coenzyme A (HMG-CoA) reductase, the rate-limiting enzyme of the L-mevalonate pathway, resulting in the reduction in intracellular cholesterol production. Many cholesterol metabolites or their nuclear receptors regulate the immune system [8], and cholesterol has been reported to contribute to sustained viral infection [9]. Moreover, statins reduce the prenylation of proteins such as Rho and Ras GTPase. The downstream signaling pathway can interfere with many steps in the viral replication cycle, including entry, replication, and spread [10].

Several recent epidemics caused by viruses in certain countries and globally represent alarming trends: the Ebola virus disease in 2014, the Middle East respiratory syndrome coronavirus (MERS-CoV) outbreak in the Middle East and South Korea, and, most importantly, the novel coronavirus disease 2019 (COVID-19) pandemic that is still ravaging the world today. Although treatments are available for specific viral infections, such as those caused by hepatitis B and C viruses, herpes virus, and human immunodeficiency virus (HIV), antiviral agents are still unavailable for many other viral infections. Moreover, new potentially pathogenic viruses may emerge in the future. Developing a specific drug for each common or emerging viral infection is challenging and not economically viable.

Although data indicate that statins possess antiviral ability, these data have been primarily obtained from in vitro studies or hospital-based observational studies [11,12,13]. To test the hypothesis that statin treatment has protective effects against viral infection in the general population, we conducted this population-based, propensity-score-matched cohort study using data from Taiwan’s National Health Insurance Research Database (NHIRD).

## 2. Methods

### 2.1. Data Sources

Implemented in 1995, Taiwan’s National Health Insurance (NHI) program is a single-payer, compulsory health insurance program that currently covers up to 99.9% of residents of Taiwan. For research purposes, the National Health Research Institutes compile medical claims in the NHI program and release the NHIRD data annually to the public. The NHIRD contains claims records of beneficiaries, including demographic data and data on prescriptions, procedures, and medications for health-care services. We analyzed data from the Longitudinal Health Insurance Database 2000, a subset of NHIRD, that contains the randomly sampled representative data of 1 million people from the registry of all beneficiaries in 2000. International Classifications of Diseases, Ninth Revision, Clinical Modification (ICD-9-CM) codes were used to define the comorbidities analyzed in this study. Data files were linked with patients’ diagnoses and replaced with surrogate numbers to ensure the confidentiality of beneficiaries. The Research Ethics Committee of China Medical University Hospital approved this study (CMUH104-REC2-115 (CR-7)).

### 2.2. Study Design and Patients

This population-based retrospective cohort study included patients with hyperlipidemia (ICD-9-CM code 272) aged ≥18 years between 2000 and 2012, and they were divided into two cohorts of 20,202 patients each: statin users and statin nonusers. The statin user cohort comprised patients who received statin therapy for at least 28 days, whereas the statin nonuser cohort comprised those without any statin therapy during the entire study period. The index date for statin users was defined as the first-time patients received statin therapy, whereas the index date for statin nonusers was randomly set. Patients with prior viral infection at the baseline period, those aged <18 years, and those with missing demographic data were excluded from the study.

Propensity score matching was performed to reduce the possibility of bias in the results and to balance the effects of potential confounders between statin users and nonusers. The propensity score was calculated using logistic regression analysis to estimate the probability of the disease status at a 1:1 ratio based on age, sex, comorbidities, and medications. Baseline comorbidities included hypertension, diabetes, rheumatoid disease, alcohol-related disease, asthma, organ transplantation, chronic liver disease, chronic kidney disease, chronic obstructive pulmonary disease, HIV, cancer, congestive heart failure, and stroke (Appendix A). Baseline medications included prednisolone, mycophenolate mofetil, cyclosporine, tacrolimus, azathioprine, thiazides, angiotensin-converting enzyme inhibitors, and angiotensin receptor blockers. The comorbidity and medication statuses were defined as their occurrence before the index date. The diagnostic accuracy of comorbidities and medications based on ICD-9 codes has been examined in previous studies [14].

### 2.3. Study Outcomes

The primary outcome of the study was the new diagnosis of viral infection, which was defined as the first diagnosis of viral infection (ICD-9-CM code 071-079, 045-049, 055-056, 060-066, and 460). To measure the incidence of viral infection, both the cohorts were followed up until the outcome occurred by the end of 2013 or their data were censored because of death or withdrawal from the insurance program, whichever occurred first. We also analyzed the incidence of viral-infection-related outcomes, including hospitalization and intubation.

### 2.4. Statistical Analyses

Age, sex, comorbidities, and medications were compared between the cohorts. The distribution of age is presented as medians with interquartile ranges (IQRs) using the Mann–Whitney U-test, and other categorical variables such as comorbidities and medications were examined using a chi-square test. A standardized mean difference statistic was used to analyze the differences between the two cohorts; values of ≤0.1 indicated negligible differences. The incidence of viral infection was calculated as the number of events divided by the sum of person-years (per 1000 person-years) for each cohort. Because patients with hyperlipidemia may not have regularly received statin therapy during the study period, an overestimation of the drug effect was possible. Therefore, to reduce any bias, statin use was set as a time-dependent covariate in the Cox proportional hazards model used to estimate hazard ratios (HRs) and 95% confidence intervals (CIs) for determining the risk of viral infection. Using this approach, patients treated with a statin were defined and included in the exposure cohort every 6 months. The patients were included in the statin nonuser cohort when they stopped receiving statin therapy during another period. The multivariate models were further adjusted for age, sex, comorbidities, and medications. The risk of viral infection was also assessed according to age, sex, and comorbidities. To explore the drugs’ effects, we evaluated the effects of the duration of statin therapy (≤110, 22–350, 351–950, and >950 days) on the risk of viral infection. We also conducted a Kaplan–Meier analysis of the cumulative incidence of viral infection in both cohorts and estimated the differences between the curves using a log-rank test. All data were analyzed using SAS (version 9.4; SAS Institute Inc., Cary, NC, USA); *p* < 0.05 was considered statistically significant.

## 3. Results

Table 1 presents a comparison of the baseline characteristics, comorbidities, and medications between the two cohorts. After propensity score matching, 20,202 patients with hyperlipidemia were included in each cohort. The median age of patients in the statin user and statin nonuser cohorts was 53.6 (IQR = 45.1–63.5) and 53.3 (IQR = 46.0–61.8) years, respectively. Both cohorts contained more men than women (approximately 55% vs. 44%). The distribution of all the covariates, including age, sex, comorbidities, and medications, was balanced between the two cohorts (standardized mean difference < 0.1).

Table 2 presents the findings of the multivariable time-dependent Cox proportional hazards regression analysis of the risk of viral infection and viral-infection-related outcomes associated with statin use. The median follow-up periods were 6.90 (IQR = 3.79–10.1) and 4.41 (IQR = 2.01–7.89) years for the statin user and nonuser cohorts, respectively.

After adjustment for confounders, the incidence rate of viral infection was significantly lower in the statin user cohort than in the statin nonuser cohort (26.3 vs. 68.1 per 1000 person-years; adjusted HR (aHR) = 0.40, 95% CI = 0.38–0.41). In addition, the rate of hospitalization owing to viral infection was significantly lower in the statin user cohort than in the statin nonuser cohort (0.27 vs. 0.73 per 1000 person-years; aHR = 0.37, 95% CI = 0.25–0.55). No significant difference was observed between the cohorts in terms of the risks of intubation due to viral infection and statin use.

Table 3 shows the results of the subgroup analyses stratified by age, sex, and comorbidities. The aHRs for the risk of viral infection were significantly lower in the statin user cohort for all age groups (*p* < 0.001). Statin use was associated with a significant decrease in the risk of viral infection in both women (aHR = 0.36, 95% CI = 0.34–0.38) and men (aHR = 0.44, 95% CI = 0.42–0.47), and in both the comorbidity (aHR = 0.41, 95% CI = 0.39–0.43) and noncomorbidity (aHR = 0.35, 95% CI = 0.31–0.38) groups.

Table 4 presents the results of the effects of longer durations of statin use on the risk of viral infection. The risk of viral infection decreased with the increased duration of statin therapy. Compared with statin nonusers, statin users had a significantly lower risk of viral infection (aHR = 0.73, 95% CI = 0.68–0.77 for ≤110 days; aHR = 0.48, 95% CI = 0.45–0.51 for 111–350 days; aHR = 0.34, 95% CI = 0.32–0.37 for 351–950 days; and aHR = 0.14, 95% CI = 0.13–0.16 for >950 days).

Kaplan–Meier analysis revealed that the cumulative incidence of viral infection was significantly lower in the statin user cohort than in the comparison cohort (log-rank test, *p* < 0.001) (Figure 1).

## 4. Discussion

In this population-based cohort study, we observed that statin users had a significantly lower risk of viral infection and viral-infection-related hospitalization than the propensity-score-matched nonusers. This protective effect was observed in both sexes and in all age subgroups of statin users; moreover, longer durations of statin therapy decreased the risk of viral infection. To the best of our knowledge, this is the first large population-based cohort study showing that statin therapy is associated with a reduced risk of viral infection.

Our findings corroborate and supplement those of numerous in vitro studies exploring the association between the use of statins and viral infections. For example, Martínez-Gutierrez et al. [15] reported that lovastatin reduces dengue virus production in epithelial and endothelial cells by inhibiting virion assembly. Moreover, simvastatin combats H1N1 infection by modulating specific cellular components in influenza A virus-infected cells [16]. Shrivastava-Ranjan et al. also revealed that lovastatin might regulate the infectivity of Ebola virus particles by interfering with glycoprotein processing [17].

Moreover, the findings of some observational studies confirmed the antiviral effects of statins. Atamna et al. found that among patients with influenza during 2017–2018, statins users were less likely to receive vasopressors and mechanical ventilation and be admitted to the intensive care unit [13]. Rodriguez-Nava1 et al. conducted a retrospective cohort study of 87 adult patients with laboratory-confirmed COVID-19 and reported that atorvastatin users showed a slower progression to death compared with nonusers [18]. Kwong et al. reported that in a cohort of >2 million adult patients aged >65 years, statin users had a significantly lower risk of influenza-related morbidity (odds ratio, 0.92; 95% CI = 0.77–0.91) [19]; however, the authors concluded that this protective effect was minimal and was perhaps caused by residual confounding. Our results are generally consistent with the findings of these studies. Moreover, to the best of our knowledge, our study is the first involving patients of all ages and both sexes, and shows that statin therapy is not only associated with a lower risk of all-cause viral infection but also a lower risk of associated hospitalization.

Although we did not explore the exact mechanisms underlying the association between statin use and the risk of viral infection in this study, previous studies have provided plausible explanations. First, as HMG-CoA reductase inhibitors, statins prevent the synthesis of mevalonate and cholesterol, both of which are crucial for sustaining the infectious viral cycle [20,21]. Second, statins inhibit the synthesis of downstream lipid isoprenoid intermediates such as geranylgeranyl pyrophosphate and farnesyl pyrophosphate, which have been reported to be associated with viral life cycle regulation and are necessary for the prenylation of several G proteins such as Ras, Rab, Rac, and Rho [12,22,23,24]. These proteins play various roles in the intracellular pathways associated with inflammation, differentiation, and proliferation. Moreover, these proteins are involved in many stages of the viral life cycle, and have been reported to function as therapeutic targets against viral infections [11,16,25,26].

The main strengths of this study were the large sample size and long follow-up period. Although the data derived from the NHI program were initially for administrative purposes, it contained detailed information regarding enrollees’ demographics, diagnoses, prescriptions, procedures, outpatient visits, and hospitalizations. Numerous observational studies have been published using NHIRD, including those associated with statins use [27,28,29].

We designed this propensity-score-matched cohort study and included many common immunosuppressive agents, various chronic diseases, organ transplantation, and HIV infection in the analysis to minimize bias caused by confounders. However, several limitations should be considered when interpreting the results. First, detailed laboratory data, including complete blood counts, differential blood counts, and hematologic malignancy staging, are unavailable in the NHIRD. Second, the detailed personal information of each enrollee was unavailable, such as alcohol consumption, diet preference, and history of exposure to hazardous substances. Third, the diagnoses and medications were based on ICD-9-CM codes retrieved from the national administrative data. Enrollees who did not seek medical care were not included in the study. This implies that socioeconomic status and access to drug treatment might not have been equitable between groups. Nevertheless, given the high accessibility and availability of medical care within the NHI system, we think that this problem is controllable. Fourth, we only included enrollees with hyperlipidemia to assess whether statin therapy alters the risk of viral infection. Whether similar results occur in the general population merits further investigation. Finally, the NHI program does not entirely cover all virus-related vaccinations, including those for influenza, human papillomavirus, and herpes zoster. We did not consider these as covariates in our analyses. It should be noted that vaccination could be more common in more complex or severely ill patients, such as in those receiving statin therapy. However, given the large-scale, long follow-up duration in this study and the accuracy of diagnosis, we think that the association between the use of statins and the risk of viral infection is valid.

In summary, this large-scale, propensity-score-matched cohort study demonstrated that statin therapy might be associated with a lower risk of viral infection in patients with hyperlipidemia. However, it should be noted that the study design was observational. The mechanisms by which statins act as antiviral agents were not explored and were inconclusive. Further evidence from blinded randomized controlled trials is warranted to facilitate relevant changes in clinical practice.

## Figures and Tables

**Figure 1 jcm-11-05626-f001:**
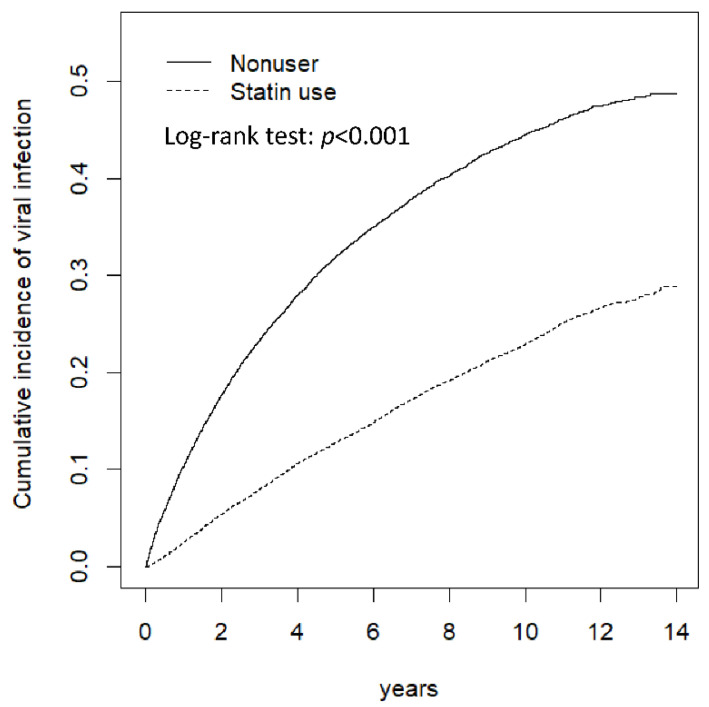
Cumulative incidence of viral infection between statin uses and nonusers.

**Table 1 jcm-11-05626-t001:** Demographic characteristics and comorbidities compared between statin users and propensity-score-matched nonusers.

	Propensity Score Matched		
Variable	Statin		Standardized Mean Differences ^§^
	No	Yes	
	N = 20,202	N = 20,202	
Age, years			
≤49	7760 (38.4)	7608 (37.7)	0.02
50–64	7957 (39.4)	8790 (43.5)	0.08
65+	4485 (22.2)	3804 (18.8)	0.08
Median ± (IQR)	53.6 (45.1–63.5)	53.3 (46.0–61.8)	0.02
Sex			
Female	8959 (44.4)	8921 (44.2)	0.004
Male	11,243 (55.7)	11,281 (55.8)	0.004
Comorbidity			
Hypertension	13,677 (67.7)	13,255 (65.6)	0.04
Diabetes	3644 (18.0)	3844 (19.0)	0.03
Rheumatoid disease	55 (0.27)	43 (0.21)	0.01
Alcohol-related disease	1828 (9.05)	1786 (8.84)	0.01
Asthma	1766 (8.74)	1780 (8.81)	0.81
Transplantation	24 (0.12)	20 (0.10)	0.01
Chronic liver disease	6056 (30.0)	6266 (31.0)	0.02
CKD or ESRD	1314 (6.50)	1398 (6.92)	0.02
COPD	2627 (13.0)	2662 (13.2)	0.01
HIV	12 (0.06)	16 (0.08)	0.01
Cancer	1297 (6.42)	1317 (6.52)	0.004
CHF	1179 (5.84)	1192 (5.90)	0.003
Stroke	1700 (8.42)	2014 (9.97)	0.001
Medications			
Prednisolone	14,248 (70.5)	14,171 (70.2)	0.008
Mycophenolate mofetil	27 (0.13)	25 (0.12)	0.003
Cyclosporine	43 (0.21)	43 (0.21)	0.000
Tacrolimus	21 (0.10)	20 (0.10)	0.002
Azathioprine	73 (0.36)	71 (0.35)	0.002
Thiazides	9122 (45.2)	9075 (44.9)	0.005
ACEI	8923 (44.2)	8837 (43.7)	0.009
ARB	8370 (41.4)	8540 (42.3)	0.017

^§^ A standardized mean difference of ≤0.1 indicates a negligible difference between the two cohorts. Abbreviations: ACEI, angiotensin II converting enzyme inhibitor; ARB, angiotensin receptor blocker; CKD, chronic kidney disease; CHF, congestive heart failure; COPD, chronic obstructive pulmonary disease; ESRD, end-stage renal disease; HIV, human immunodeficiency virus; IQR, interquartile range.

**Table 2 jcm-11-05626-t002:** Incidence (per 1000 person-years) of viral infection and the estimated Cox proportional hazards ratios of statin users to nonusers based on time-dependent exposure covariates in patients with hyperlipidemia.

	Propensity Score Matched	
	Statin	
	No	Yes
Variables	(N = 20,202)	(N = 20,202)
Person-years	104,665	141,459
Follow-up time (y), Median ± (IQR)	4.41 (2.01–7.89)	6.90 (3.79–10.1)
Viral infection		
Event	7125	3723
Rate ^#^	68.1	26.3
Crude HR (95% CI)	1 (Reference)	0.41 (0.39, 0.42) ***
Adjusted HR ^†^ (95% CI)	1 (Reference)	0.40 (0.38, 0.41) ***
Hospitalization due to viral infection		
Event	76	38
Rate ^#^	0.73	0.27
Crude HR (95% CI)	1 (Reference)	0.38 (0.26, 0.56) ***
Adjusted HR ^†^ (95% CI)	1 (Reference)	0.37 (0.25, 0.55) ***
Intubation due to viral infection		
Event	3	3
Rate ^#^	0.03	0.02
Crude HR (95% CI)	1 (Reference)	0.72 (0.14, 3.60)
Adjusted HR ^†^ (95% CI)	1 (Reference)	0.39 (0.06, 2.41)

Rate ^#^, incidence rate per 1000 person-years; crude HR, relative; adjusted HR ^†^, multivariable analysis including age, sex, comorbidities, and medications. Abbreviations: CI, confidence interval; HR, hazard ratio; IQR, interquartile range. *** *p* < 0.001.

**Table 3 jcm-11-05626-t003:** Incidence and HRs of viral infection by age group and sex in patients with hyperlipidemia.

Statin						
	No (N = 20,202)		Yes (N = 20,202)			
Variables	Event	Rate ^#^	Event	Rate ^#^	Crude HR (95% CI)	Adjusted HR ^†^ (95% CI)
Age, years						
≤49	2903	68.8	1421	25.1	0.38 (0.36, 0.41) ***	0.38 (0.36, 0.40) ***
50–64	2816	70.0	1645	27.6	0.41 (0.39, 0.44) ***	0.40 (0.38, 0.43) ***
65+	1406	63.1	657	26.0	0.44 (0.40, 0.48) ***	0.43 (0.39, 0.47) ***
Sex						
Female	3680	82.0	1741	27.7	0.36 (0.34, 0.38) ***	0.36 (0.34, 0.38) ***
Male	3445	57.6	1982	25.3	0.46 (0.43, 0.48) ***	0.44 (0.42, 0.47) ***
Comorbidity ^§^						
No	1332	97.1	580	31.5	0.35 (0.31, 0.38) ***	0.35 (0.31, 0.38) ***
Yes	5793	63.7	3143	25.6	0.42 (0.40, 0.44) ***	0.41 (0.39, 0.43) ***

Rate ^#^, incidence rate per 1000 person-years; crude HR, relative; adjusted HR ^†^, multivariable analysis including age, sex, comorbidities, and medications. ^§^ Individuals with any comorbidity, including hypertension, diabetes, rheumatoid disease, alcohol-related disease, asthma, transplantation, chronic liver disease, chronic kidney disease or end-stage renal disease, chronic obstructive pulmonary disease, human immunodeficiency virus, cancer, congestive heart failure, and stroke, were classified into the comorbidity group. Abbreviations: CI, confidence interval; HR, hazard ratio. *** *p* < 0.001.

**Table 4 jcm-11-05626-t004:** Incidence and adjusted hazard ratio of viral infection stratified by the duration of statin therapy in patients with hyperlipidemia.

Medication Exposed	N	Event	Person-Year	Rate	Adjusted HR (95% CI) ^†^
Statin ^#^					
Non-statin	20,202	7125	104,665	68.1	1.00
≤110 days	3968	1250	23,638	52.9	0.73 (0.68, 0.77) ***
111–350 days	6059	1288	37,958	33.9	0.48 (0.45, 0.51) ***
350–950 days	5246	811	36,122	22.5	0.34 (0.32, 0.37) ***
>950 days	4929	374	43,740	8.55	0.14 (0.13, 0.16) ***

Rate, incidence rate per 1000 person-years. ^#^ Cumulative use days are partitioned into four segments by quartile. Adjusted HR ^†^: multivariable analysis including age, sex, comorbidities, and medications. Abbreviations: CI, confidence interval; HR, hazard ratio. *** *p* < 0.001.

## Data Availability

The dataset used in this study is held by the Taiwan Ministry of Health and Welfare (MOHW). The Ministry of Health and Welfare must approve our application to access this data. Any researcher interested in accessing this dataset can submit an application form to the Ministry of Health and Welfare requesting access. Please contact the staff of MOHW (Email: stcarolwu@mohw.gov.tw) for further assistance. Taiwan Ministry of Health and Welfare Address: No.488, Sec. 6, Zhongxiao E. Rd., Nangang Dist., Taipei City 115, Taiwan (R.O.C.). Phone: +886-2-8590-6848.

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
