# Peer review of "Statin Therapy and the Risk of Viral Infection: A Retrospective Population-Based Cohort Study"

_jcm, 2022, doi:10.3390/jcm11195626_

Round 1

Reviewer 1 Report

A well performed population-based study to test the risk of viral infections between statins users and nonusers, in Taiwan population, through the NHI database. In general the development and presentation is appropriate.

A few issues should be consider and solve if required:

.- the expression 0.4-fold lower could be doubtful, perhaps a more direct style as the risk of... is 0.4 (IC...) could be more appropriate

.- For this kind of study, control of bias and confounders is critical, to avoid misinterpretation or spurious relationship. Please justify, or comment in limitations, if the information bias (i.e. the chance to record the information regarding the risk or the exposure is similar in both groups; this could be affected e.g. if the access to drug treatments were non equitative, or huge social differences could exist between groups) is adequately controlled.

.- Justify if it is considered and analysed the existence of virus vaccination in both groups. Otherwise it could appear that for more complex or severely ill patients (those receiving lipid lowering drugs), the vaccination could be more common.

.- To avoid misinterpretation, and given that the mechanism is not being studied, it should be remarked that given the design of the study, it is not the capability of statins to act as antivirals what is being checked, but the risk of being attended by a virus infection between the exposure groups.

Reviewer 2 Report

Dear Editor,

I carefully read the manuscript by Wu et al.

My comments and suggestions for the authors are the following:

 - The abstract should be formatted following the Instructions for the Authors in one single paragraph.

 - The main strength of the study is its large sample size, of course. I suggest the authors to discuss further on this point.

 - The authors should more deeply discuss on the limitations of their analisis, particularly in light of the observations raised by Cicero et al. in PMID: 32768364.

- Statistical analysis should be more properly described. I also encourage the authors to perform a post-hoc power analysis.

 - The authors wrote that "Baseline comorbidities included hypertension, diabetes, rheumatoid disease, alcohol-related disease, asthma, organ transplantation, chronic liver disease, chronic kidney disease, chronic obstructive pulmonary disease, HIV, cancer, congestive heart failure, and stroke. Baseline medications included prednisolone, mycophenolate mofetil, cyclosporine, tacrolimus, azathioprine, thiazides, angiotensin-converting enzyme inhibitors, and angiotensin receptor blockers". Howeber, the authors should specify how these pieces of information were collected. Did they use standardized questionnaires? Which guidelines did they refer to in the diagnosis of hypertension and diabetes?
